# On Adversarial Attacks In Acoustic Localization

**Tamir Shor**                                                    *tamir.shor@campus.technion.ac.il*
*Department of Computer Science*
*Technion – Israel Institute of Technology*
*Haifa, Israel*

**Chaim Baskin**                                                    *chaimbaskin@bgu.ac.il*
*School of Electrical and Computer Engineering*
*Ben-Gurion University of the Negev*
*Be'er Sheva, Israel*

**Alex Bronstein**                                                    *bron@cs.technion.ac.il*
*Technion – Israel Institute of Technology*
*Haifa, Israel*
*Institute of Science and Technology Austria*

**Reviewed on OpenReview:** *https://openreview.net/forum?id=Nxm5xXoLFb*

## Abstract

Multi-rotor aerial vehicles (drones) are increasingly deployed across diverse domains, where accurate navigation is critical. The limitations of vision-based methods under poor lighting and occlusions have driven growing interest in acoustic sensing as an alternative. However, the security of acoustic-based localization has not been examined. Adversarial attacks pose a serious threat, potentially leading to mission-critical failures and safety risks. While prior research has explored adversarial attacks on vision-based systems, no work has addressed the acoustic setting. In this paper, we present the first comprehensive study of adversarial robustness in acoustic drone localization. We formulate white-box projected gradient descent (PGD) attacks from an external sound source and show their significant impact on localization accuracy. Furthermore, we propose a novel defense algorithm based on rotor phase modulation, capable of effectively recovering clean signals and mitigating adversarial degradation. Our results highlight both the vulnerability of acoustic localization and the potential for robust defense strategies.

## 1 Introduction

Multi-rotor autonomous aerial vehicles (drones) have seen rapid adoption across a wide range of industries (Ayamga et al., 2021; Merkert & Bushell, 2020; Moshref-Javadi & Winkenbach, 2021), including emergency medicine (Johnson et al., 2021; Zailani et al., 2020), sustainability (Dutta & Goswami, 2020; Mahroof et al., 2021), and disaster control (Daud et al., 2022; Ishiwatari, 2024), as well as recreational and commercial applications (Tan et al., 2021; Benarbia & Kyamakya, 2021). Their popularity stems from the ability to operate in challenging or hazardous environments while reducing human risk and cost.

Successful drone deployment relies on accurate and efficient navigation. This has driven major advances in localization methods (Kang et al., 2015; Dijkshoorn, 2012; Sorbelli et al., 2018), particularly with deep learning (Yousaf et al., 2022; Bisio et al., 2021; Zhang et al., 2022). Current solutions primarily depend on GPS with inertial systems, visual odometry, or active sensors such as LiDAR (Arafat et al., 2023; Aburaya et al., 2024; Niu et al., 2022; Debeunne & Vivet, 2020). While effective in many scenarios, these methods often fail under degraded lighting, occlusions, or structural constraints, and may be impractical in GPS-denied environments (Arafat et al., 2023; Al-Radaideh & Sun, 2021; Dreissig et al., 2023; Meles et al., 2023). To address these limitations, researchers have explored alternative modalities, notably *acoustic localization* (He

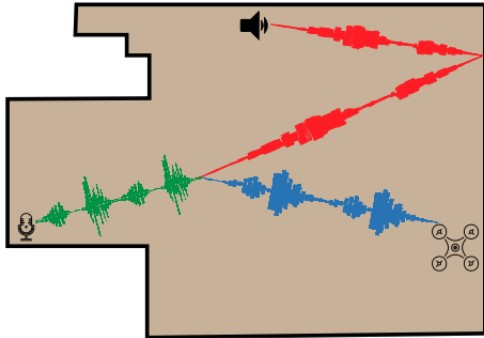

Figure 1: **Adversarial Acoustic Localization Setting -**. Localization model input (green) is the drone sound response (blue) perturbed with an external speaker adversarial interference (red).

et al., 2023; Sun et al., 2023; Serussi et al., 2024), which leverages sound propagation for robust localization even under visual or RF impairments (Famili et al., 2022). With the growing reliance on drones, security concerns intensify. Adversarial attacks have been shown to manipulate perception systems, leading to severe outcomes such as navigation failure or collisions (Mynuddin et al., 2023; Wisniewski et al., 2024; Guesmi & Shafique, 2024; Fu et al., 2021). While defenses against such attacks exist (Mynuddin et al., 2024; Wang et al., 2023), they have focused exclusively on visual or LiDAR-based sensing. Despite the rapid adoption of acoustic localization, its adversarial robustness remains unexplored.

To address this gap, we present the first formulation and analysis of adversarial attacks targeting acoustic drone localization. We study white-box attacks from a single omni-directional sound source external to the drone. Additionally, we leverage the phase modulation mechanism proposed in Serussi et al. (2024) to design a novel defense method that separates the clean signal from adversarial interference under minimal assumptions. We further extend Serussi et al. (2024) to real-world acoustic recordings, beyond simulated settings, enabling realistic evaluation. While our experiments focus on drone navigation, the proposed approach generalizes to any single-agent acoustic localization system.

Our main contributions are:

1. We formulate and benchmark adversarial attacks on acoustic localization, with a fully differentiable attack pipeline.

2. We analyze the computational and performance implications of perturbation source-location optimization.

3. We develop an acoustic-channel attack test-time defense algorithm, capable of credibly reconstructing the original perturbation waveform sampled by the drone.

4. We extend Serussi et al. (2024) to demonstrate self-sound-based localization using real acoustic data.

## 2 Related Work

In this work we adopt the acoustic localization algorithm from Serussi et al. (2024). This algorithm performs drone localization based solely on the self-sound emitted by the drone's propulsion system, as sampled by a circular array of microphones $\mathcal{M}$ located around the drone. The authors propose a three-step pipeline. First, a *forward model* is used to model the self-sound emitted by the drone's propulsion system in free space, using a fixed, parametrized set $\mathcal{S}$ of point sound sources optimized to fit an actual free-space recording. Second, a neural, transformer-based *inverse model* is used to regress the drone location based on the propulsion sound as sampled by the microphone array. This sound is simulated using both the forward model and RIR by superimposing the direct path from each point sound source to each microphone, along with the reflected paths from the walls. These paths are given by the image source model (ISM), replacing source reflections

on walls with imaginary point sound sources (termed *images* - Allen & Berkley (1979)). Lastly, the authors present the optimization of the time-dependent angular offsets of the rotors, termed *phase modulation*, in order to improve localization accuracy. We further elaborate on the concept of phase modulation in Section 3.4 in the sequel. Our reason for using the approach from Serussi et al. (2024) is the fact that it presents an accurate, purely acoustic-based localization model for us to evaluate under acoustic adversarial perturbations, and mainly since the proposed phase modulation mechanism will serve us in establishing our proposed defense method in Section 3.4.

One shortcoming of Serussi et al. (2024) is the choice to compute input to the location regressor (a.k.a inverse model), i.e. the sound sampled at each sensor using a simulation superimposing a set of point sources given by the ISM, with equation 1 being applied over each point sound source separately. While this method is relatively accurate in sufficient point source sample density (Scheibler et al., 2018), the computational costs entailed in computing the ISM render this method prohibitively expensive for larger and more complex acoustic environments. Our specific use-case of adversarial attack optimization calls for the optimization of sound emitted by the attacker (Section 3.3), obligating rapid inference of the forward model (i.e., through sound generation). This makes the original ISM-based approach of sound computation inapplicable for our purposes. In this work we therefore opt develop a modified version of the ISM-reliant algorithm from Serussi et al. (2024), making use of neural-acoustic fields (NAFs) (Luo et al., 2022), as further elaborated on Section 3.2.

## 3 Method

In the following section, we outline our approach to developing adversarial attacks and defenses for the task of acoustic drone localization. Section 3.1 lays the foundations of acoustic sensing-guided localization, and Section 3.2 presents the "clean" (non-perturbed) localization setting studied in this paper. Section 3.3 formulates our threat model and the proposed acoustic adversarial attack, and Section 3.4 proposes a defense method relying on active rotor phase modulation.

### 3.1 Acoustic Localization Background

Acoustic localization is the task of identifying the location $\mathbf{x}_{\mathrm{s}} \in \mathbb{R}^n$ of a sound source emitting a sound signal $s_{\mathrm{s}}(t) \in \mathbb{R}^T$ in a known environment, based on the response of that signal $s_{\mathrm{m}}(t) \in \mathbb{R}^T$ as measured at a given sensor (microphone) location $\mathbf{x}_{\mathrm{m}} \in \mathbb{R}^n$. Here, the temporal signals are assumed to be sampled at $T$ discrete time points, and $n$ denotes the number of location degrees of freedom. While, in the most general case, $n = 6$ (for location and orientation in 3D space), in this work, we focus on 2D localization with known orientation ($n = 2$). State-of-the-art acoustic localization solutions (Baron et al., 2019; Serussi et al., 2024) typically employ various neural architectures $\mathcal{F}(s_{\mathrm{m}}(t))$ optimized to regress $\mathbf{x}_{\mathrm{s}}$.

$s_{\mathrm{m}}(t)$ is dependent on the decay and scatter $s_{\mathrm{s}}(t)$ undergoes when propagated through the room to $\mathbf{x}_{\mathrm{m}}$. This propagation is modeled by the room impulse response (RIR) (Borish, 1984) $r(t; \mathbf{x}_{\mathrm{s}}, \mathbf{x}_{\mathrm{m}}, \zeta)$ – a temporal function also dependent on the source and sensor locations, as well the room geometry and physical properties, collectively denoted as $\zeta$. This function essentially describes the response in time of an impulse at $\mathbf{x}_{\mathrm{s}}$ as perceived at $\mathbf{x}_{\mathrm{m}}$ after propagating in the environment $\zeta$. The sampled sound $s_{\mathrm{m}}$ can finally be attained by convolving the RIR with the input sound,

$$s_{\mathrm{m}}(t) = r(t; \mathbf{x}_{\mathrm{s}}, \mathbf{x}_{\mathrm{m}}, \zeta) * s_{\mathrm{s}}(t), \tag{1}$$

where the convolution is performed over time $t$.

### 3.2 Clean localization

To circumvent the reliance on computationally complex ISMs used in Serussi et al. (2024) for RIR computation (as detailed in Section 2), we replace usage of ISMs with neural acoustic fields (NAFs, Luo et al. (2022)). NAFs offer a reliable neural representation of the RIR, and are trained over actual RIR sampled from actual room acoustics. Unlike the original forward model from Serussi et al. (2024), NAFs produce RIRs already

encompassing higher-order reflection sources. Therefore, for every point sound source, modeling the drone sound $s_{\mathrm{s}}$, and the entire sound waveform sampled at a given microphone, $s_{\mathrm{m}}$, can be directly derived from equation 1 in a single forward pass, with the RIR component attained by a single query of the NAF model.

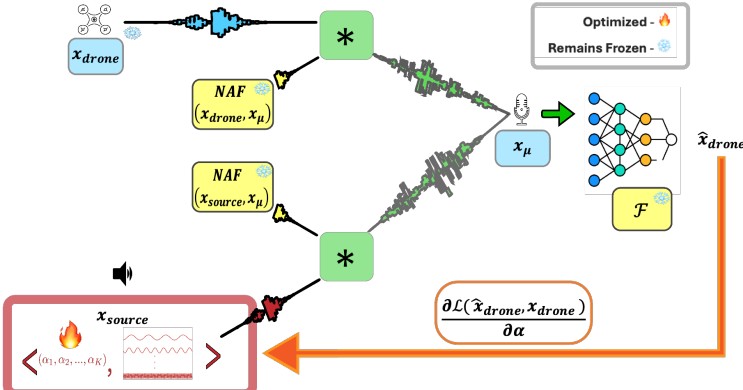

Figure 2: **Adversarial Pipeline Overview -** sound from both drone sound and adversarial source is convolved with NAF-induced RIRs and superimposed at the sensor, prior to being fed into the localization model. Localization loss gradients guide the optimization of source-emitted waveforms.

In our altered formulation for Serussi et al. (2024), for every microphone $\mu \in \mathcal{M}$, the response of each point sound-source $s_i \forall i \in \mathcal{S}$ as perceived at $\mu$, is computed using equation 1 (with NAF-produced RIRs). The total sampled sound at sensor $\mu$ is given by $s_\mu = \sum_i s_i$. These inputs are fed into the inverse model (transformer-encoder architecture from Serussi et al. (2024)) to train our "clean" regressor, to be later evaluated under the presence of adversarial acoustic perturbations. We report clean localization performance in Section 4.2. These modifications fall outside the main intended contributions of this paper and serve as a stepping stone toward a fast, differentiable model for the computation of $s_{\mathrm{m}}$.

### 3.3 Acoustic Adversarial Attacks

In their most general form, adversarial attacks aim to add a perturbation $p$ to the input of a given, usually trained model $\mathcal{M}(x)$ receiving an input $x$, so as to manipulate the performance of the model over the perturbed input $x+p$ in order to maximize the model's error according some quality measure $\mathcal{L}$, thus harming the model's performance and reliability. This is usually done by solving the constrained optimization problem formulated in equation 2:

$$\max_p \mathcal{L}(\mathcal{M}(x+p)) \quad \text{s.t.} \quad \mathcal{L}_q(p) \leq B \tag{2}$$

where $\mathcal{L}_q$ is some constraint over the optimized perturbation, ensuring its feasibility under domain-specific requirements.

In our setting, $\mathcal{M}$ is the localization model from Section 3.2, and x is sound sampled at the drone's microphone array. Our acoustic adversarial attack must establish both the perturbation $p$, and the a set of constraints (collectively measured by $\mathcal{L}_q$) appropriate for our domain of acoustic localization.

### 3.3.1 Perturbation Formulation

In our setting we wish to develop adversarial attacks originated from a stationary, omni-directional sound source located at an arbitrary location in the environment, controlled by the attacker. The motivation for this choice is that unlike common adversarial perturbations known from the field of Computer-Vision, such as 2D perturbations added to the input of image classifiers, in our setting the attacker cannot directly control the perturbation added to the model's input signal. This is because, unlike spatial signals (e.g. 2D images), acoustic signals are not stationary – the presence of an acoustic perturbation at one location in the environment inevitably affects the sampled sound at another location. For this reason, we also focus our discussion in this paper on universal adversarial attacks, rather than optimizing a distinct perturbation for every drone location and orientation.

We choose to model the sound emitted by the perturbation sound source (namely, the attacker) $s_{\mathrm{p}} \in \mathbb{R}^T$ using a basis $\mathcal{B} = \{f_k\}_{k \in K}$ of sine waves from a chosen frequency set $K$. $f_k \forall k \in K$ is a sine wave of frequency $k$ sampled at $T$ timesteps . The corresponding perturbation is given by spanning this basis with a set of $|K|$ learnable amplitudes $\{\alpha_k\}_{k \in K}$:

$$s_{\mathrm{p}} = \sum_{k \in K} \alpha_k \cdot f_k \tag{3}$$

The rationale for this formulation is 3-fold: Firstly, it implies inherent periodicity, as we wish to optimize for periodic perturbations to avoid having to optimize the attack for an indefinite temporal duration. Especially under the assumption that the clean signal is in itself periodic (as is the case in the sound generated by the drone). Secondly, this formulation allows to reduce the number of learnable parameters from $T$ (when optimizing for the sound in signal domain) to $|K|$, allowing more efficient optimization. Lastly, it allows simple control over the amplitude and frequency components of the perturbation in order to impose various constraints as elaborated in Section 3.3.2.

### 3.3.2 Attack Constraints

Our goal in this section is to develop a set of constraints that would allow the attack to efficiently harm localization performance while still not be easily detectable by the localizing agent. Contrary to the computer-vision domain and similar counterpart adversarial settings, where appropriate constraints are well-studied and well-established (see Croce et al. (2020)) (mostly $L_\infty, L_2$ and $L_1$ norms, among others), the relatively under-explored domain of acoustic attacks does not currently hold a set of widely accepted attack constraints. We deem the acoustic setting more challenging in this aspect as, unlike vision where many scenarios may bound together under relation with human-perception, the conditions where an acoustic perturbation can be deemed feasible or non-trivial for detection vastly change under different factors (e.g. sensors, "clean" sound properties and task). For this reason, we propose our own constraints for the optimized adversarial perturbation. The effect of different constraint value choices over performance is studied in Section 4.2.

**Frequency constraints.** Our first observation in this context is that the sound emitted by the drone's propulsion system is periodic. If the period of perturbative sound substantially differs from that of the drone, the adversarial attack could be trivially detected by the agent. Therefore, we filter our sine basis $\mathcal{B}$ to include only frequencies integerly intertwined in the drone's cycle. The bandwidth of perturbation source-emitted frequencies is also subjected to frequency constraints given by the sound-source itself. We, therefore, also clamp the bandwidth of $\mathcal{B}$ according to some plausible minimal and maximal values (set to a minimum of 50 Hz and a maximum of 2 KHz in all of our experiments). All frequency-related constraints are imposed by the construction of the perturbation as detailed in Section 3.3.1.

**Signal constraints.** We constrain both the amplitude and power of the emitted signal. The amplitude is constrained to avoid trivial perturbations that dominate the clean signal and can subsequently be easily disentangled from it. The power is more closely related to human perception (namely the loudness of perceived signal), however constraining it also helps diminish irregular local signal energy patterns that may potentially be utilized in attack detection.

Signal constraints are imposed via soft constraints regularizing the optimization process (equation 4). Note that while the perturbation is sampled at the microphones, all constraints are imposed over the sound at the source. This choice is motivated by the fact that the sound sampled at the microphones could potentially greatly vary with drone spatial translation, reducing attacker control over perturbation feasibility.

**Location constraints.** Upon optimizing for the perturbative source location $\mathbf{x}_{\mathrm{p}}$, we must ensure the source remains within environment boundaries. This is done using a signed-distance-function (SDF) $\mathcal{L}_{SDF}$ linearly penalizing distance when the source is optimized to be away from environment boundaries.

### 3.3.3 Perturbation Propagation

Much like the "clean" sound emitted by the drone, our adversarial perturbation must also be propagated to the drone's sensor array to be sampled by it. Our goal is to formulate a universal attack optimization scheme shaping the emitted signal at the source $s_{\mathrm{p}}$ so that the sampled response at the sensor $\mu$, denoted $\sigma_{\mathrm{p}}$, would

optimally reduce localization accuracy. In our experiments, we consider both optimization of the emitted perturbation for a sound source located in the center of the room, as well as optimization of the emitted sound jointly with the perturbation speaker's location $\mathbf{x}_\mathrm{p} \in \mathbb{R}^2$. For our localization quality measure, we choose Mean-Squared-Error (MSE), following the criterion from Serussi et al. (2024). Our universal attack optimization, in its most general form, is therefore formulated according to equation 4:

$$\max_{\{a_k\}_{k \in K}, \mathbf{x}_\mathrm{p}} \|\mathcal{F}(\{r(t; \mathbf{x}_\mathrm{p}, \mathbf{x}_\mu) * s_\mathrm{p}(t)\}_{\mu \in \mathcal{M}}) - \mathbf{x}_\mathrm{d}\|_2^2 + \mathcal{L}_q(s_p, x_p) \tag{4}$$

for definition of the constraint loss $\mathcal{L}_q$ as:

$$\mathcal{L}_q(s_p, x_p) = \lambda_{amp} \cdot \max\{0, \|s_\mathrm{p}\|_\infty - \beta\}$$
$$+ \lambda_{power} \cdot \max\left\{0, \sum_{t=1}^{T} s_\mathrm{p}(t)^2 - \gamma\right\} + \lambda_{SDF}\mathcal{L}_{SDF}(\mathbf{x}_\mathrm{p})$$

$s_\mathrm{p}$ is the perturbation source-sound attained from equation 3, $r(t; \mathbf{x}_\mathrm{p}, \mathbf{x}_\mu)$ is the RIR between sound-source location $\mathbf{x}_\mathrm{p}$ and microphone location $\mathbf{x}_\mu$, given by the pre-trained NAF, $\mathcal{F}$ is the pre-trained localization model with a set of microphones $\mathcal{M}$, where each microphone $\mu$ is located in location $\mathbf{x}_\mu$. $\mathbf{x}_\mathrm{d}$ represents actual location of the center of the drone, and $\beta, \gamma$ are the respective amplitude and power bounds from Section 3.3.2, for losses weighted by $\lambda_{amp}$ and $\lambda_{power}$. Location SDF is weighted by $\lambda_{SDF}$. While equation 4 is formulated for non-targeted attacks, in Section 4.2 and in Appendix B.1 we adapt our formulation to explore targeted attacks.

### 3.3.4 Source Location Optimization

One potential difficulty in our formulation from equation 4 is that there are cases where optimization of the perturbation source location $\mathbf{x}_\mathrm{p}$ within the scene may be difficult or altogether impossible. The reason is that optimizing $\mathbf{x}_\mathrm{p}$ necessitates differentiation through the acoustic model of the environment (namely, in our case, the NAF). To address cases where such a differentiable model is not available or where the added computational costs incurred by querying and backpropagating through such, usually large, models are too heavy (see Appendix D for time and memory footprint in our experiments), in our experiments, we also analyze universal attacks with a fixed perturbation source location in the center of the room. In this case we only optimize the perturbative sound. The alleviation in computational costs stems from the fact that the perturbation impulse response for every microphone location is constant, circumventing repeated forward and backward evaluation of the NAF model. As Section 4.2 shows, we observe only a marginal decrease in attack prowess when waiving source-location optimization. Our adversarial pipeline is illustrated in Figure 2.

### 3.4 Phase Modulation Perturbation Delineation

In this section, we propose Phase Modulation Perturbation Delineation - a novel method for the utilization of phase modulation (Serussi et al., 2024) for the recovery of the "clean," non-perturbed signal from the entire (presumably adversarially perturbed) acoustic sample as perceived by any of the drone's sensors. Section 3.4.1 overviews the concept of phase modulation, originally used by Serussi et al. (2024) for improving localization accuracy. In Section 3.4.2, we show how this mechanism can be further extended for perturbation and clean-signal separation.

### 3.4.1 Phase Modulation

In our acoustic localization setting, similar to that studied in Serussi et al. (2024), the drone's propulsion system consists of a set of $\mathcal{R}$ rotors that rotate at constant and equal angular velocities. The resulting angular locations of each rotor through the duration of a single cycle spanning over $T$ seconds (namely, the azimuthal rotor shaft positions w.r.t some chosen starting point) can thus be described as a function $\varphi_0(t) : [T] \to [0, 2\pi]$ converting moments in time from each cycle to the corresponding angular location of

rotor $i$.

The phase modulation mechanism from Serussi et al. (2024) proposes to actively optimize a per-rotor function of temporal offsets from $\varphi_0(t)$, termed a *modulation function $\varphi_i(t) : [T] \to [0, 2\pi]$*, for every rotor $i$. Every combination of such modulation functions alters the sound emitted by the propulsion system through time, and thus the rationale is to optimize the set of $\mathcal{R}$ such per-rotor modulation functions to generate input waveforms more useful for localization (under imposition of certain constraints over these modulation functions, ensuring kinematic feasibility and the drone's stable flight).

### 3.4.2 Perturbation Delineation

We extend the concept of phase modulation (Section 3.4.1) to delineate the adversarial perturbation from the entire sampled input $s_\mu$ at a given microphone $\mu$. Let us denote the perturbation response of $s_{\mathrm{p}}$ at microphone $\mu$ by $\sigma_{\mathrm{p}}$. Upon sampling a perturbed waveform $s_\mu = s_{drone} + \sigma_{\mathrm{p}}$, the goal is to solve for $\sigma_{\mathrm{p}}$. Upon doing so, we can input the original non-perturbed signal $s_{drone}$ to the localization model, essentially nullifying the effect of the attack over localization performance.

Our approach is applying gradual temporal delays over $s_{drone}$ using the phase modulation mechanism, utilizing the fact the perturbation sound remains constant under these modulations. Denote by $T$ the maximum of the period times of the drone and perturbation source. Since the drone cycle $T_{drone}$ must be an integer multiple of the perturbation source period $T_{pert}$ (Section 3.3.2), both $s_{drone}$ and $s_{\mathrm{p}}$ complete a periodic cycle (of one or more periods) within a duration of $T$ seconds.

For every moment in time $j \in [T]$, we can apply constant phase modulation of $j$ timesteps across all rotors jointly for the entire $T$ seconds period. Denote the sound sampled at some arbitrary microphone $\mu$ with constant modulation $j$ across rotors as $s_\mu(t; j)$. This sound is the sum of the $j$-modulated drone-emitted sound, denoted $s_{drone}(t; j)$, and the original sampled perturbation sound $\sigma_{\mathrm{p}}(t)$ (unaffected by phase modulation). Our key observation is that for every non-zero value of $j$, at moment $j$ of the current period (where each rotor is modulated at $j$ timesteps), the sound component originated from the drone's propulsion system is the same as that sampled at the $j = 0$ constant modulation at timestep 0, and hence by simple subtraction: $s_\mu(t = j; j) - s_\mu(t = 0; 0) = \sigma_{\mathrm{p}}(t = j) - \sigma_{\mathrm{p}}(t = 0)$.

By performing this process for every value of $j \in [T]$, we can essentially recover $\sigma_{\mathrm{p}}(t) - \sigma_{\mathrm{p}}(t = 0)$ for every value of $t$, giving us the perturbating waveform at the location of the microphone $\mu$. One degree of uncertainty which our algorithm does not allow solving for, is the value of $\sigma_{\mathrm{p}}$ at moment $t = 0$. While we believe this value can be retrieved, for example by imposing correlation among several perturbation waveforms at different locations, we defer the study of this possibility to future work. For the scope of this work, as validated in Section 4.3, simple setting of $\sigma_{\mathrm{p}}(t = 0)$ to be 0 everywhere in the environment reduces localization degradation almost completely. We stress that while our proposed algorithm is demonstrated on phase modulation of drone rotors, it is applicable for acoustic perturbation delineation of any agent capable of actively shaping the "clean" sound, thus separating it from the constant perturbation. This formulation is summarized in algorithm 1. Figure 3 visually depicts our algorithm - a single drone period is illustrated

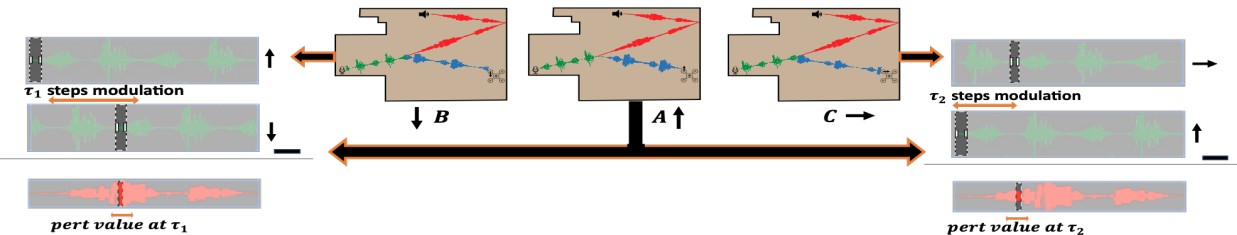

Figure 3: **Summary of Perturbation Delineation Method -** rotor self-sound (blue) under different phase modulations (black arrows depict rotor angular location at $t = 0$). In red is the perturbation sound. The two are superimposed to the microphone-sampled signal (green).

under 3 different phase modulations - 0 (A), $\pi$ (B, corresponding to $\tau_1$ timesteps), and $\frac{pi}{2}$ (C, corresponding to $\tau_2$ timesteps). In each room the same perturbation sound (red) is observed, while for each modulation the same drone-emitted sound is observed up to a certain shift (blue). The superimposed sound sampled by

the microphone (green) is shifted accordingly (w.r.t the 0-modulation room A) in each room. In the bottom part, we illustrate how the 0-modulation waveform is subtracted from each $\tau$-step modulated waveform to receive the original perturbation (red waveform) at timestep $\tau$, utilizing the fact the perturbation is agnostic to phase modulation.

---

**Algorithm 1** Perturbation Recovery Algorithm for Sound Waveform Separation

---

**Require:** Perturbed waveform $s_\mu$, period time $T$
**Ensure:** Recovered sampled perturbation waveform $s_r$

  1: Initialize $s_r = [\ ]$                                                   ▷ Recovered waveform
  2: **for** $j = 1$ **to** $T$ **do**
  3:       sample $s_\mu(t; j)$                                       ▷ Modulate by $j$ timesteps
  4:       update $s_r(t = j) = s_\mu(t; j) - s_\mu(t; 0)$            ▷ Perturbation at timestep $j$
  5: **end for**
  6: **return** $s_r$

---

## 4 Results

### 4.1 Experimental Setup

Following the experimental setting in Luo et al. (2022), we pick a representative subset of acoustic environments from the Matterport3D (Chang et al., 2017) and Replica (Straub et al., 2019) datasets used in training of RIRs from Neural Acoustic Fields (NAFs). In this section, we primarily present results from the *apartment_2_frl* environment (denoted *apt*, in this section). We also provide partial results for the *office_4* and *room_2* scenes in Figure 5, with the full set of results available in the supplementary material (Appendix B). For each environment we train the clean localization model from Serussi et al. (2024), using the environment-fitted pre-trained NAF. In order to explore our developed approaches disjointly from any external, case-specific affects, in most experiments we assume zero interference from any acoustic signals, excluding the drone and the attacker. Nonetheless, in Section 4.4 we also model the effect of white noise over adversarial success. Furthermore, in our experiments we focus on the localization model from Serussi et al. (2024) since it is the only one completely reliant on the self-sound emitted by the drone's propulsion system. Other methods (e.g. Sun et al. (2023); He et al. (2023)) rely mostly on external speakers, and thus require the development of adversarial attacks accounting for both the speaker sound and propulsion sound simultaneously. We refer this direction of research to future work. Nonetheless, we emphasize that our developed attack and defense methods are applicable to any combination of drone configuration and acoustic localization algorithm. Localization training details are stated in Appendix C.

To evaluate our attacks, for every combination of constraints $(\beta, \gamma) \in [0.01, 0.1, 0.5, 1] \times [0.1, 0.25, 0.5, 1, 2]$ we perform 100 iterations of universal PGD attack over each environment and combination, with early-stop of 5 iterations without localization loss increase. Our set of attack constraints is upper-bounded to not surpass 50% of the average clean signal amplitude and power, and lower-bounded to allow minimal deviation from clean localization. We set all regularization weights from our training objective (equation 4) to 1 in all experiments. We report our results in scaled RMS (i.e. RMS between predicted and ground-truth locations after being scaled to the $[0, 1]$ range) for the sake of interpretability and comparability across scenes.

### 4.2 Attack Results

Optimized attack RMS error results across different bounds are reported in Figure 4. Our proposed attack increases the mean localization error from slightly below 5% in the clean model, to 37.4% for highest amplitude and power bounds. Results also demonstrate saturation of attack efficacy with growth in the $\beta, \gamma$ bounds, where in both figures the largest and second-largest amplitude bounds $\beta = 0.5, \beta = 1$ intersect almost completely. This is also true for the power bounds, as all figures remain almost constant in transition from $\gamma = 1$ to $\gamma = 2$. This bounds the maximum efficacy of our PGD attack in bound of around 0.5 for the amplitude and $\gamma = 1$ for power. Lastly, corresponding to Section 3.3.4, we observe that results for location optimization (left) and fixed-source location experiments (right) are nearly identical. We deem the apparent

agnostic nature of sound-source perturbation optimization to source location a notable conclusion for future related studies, especially given the increased computational costs (see Appendix D).

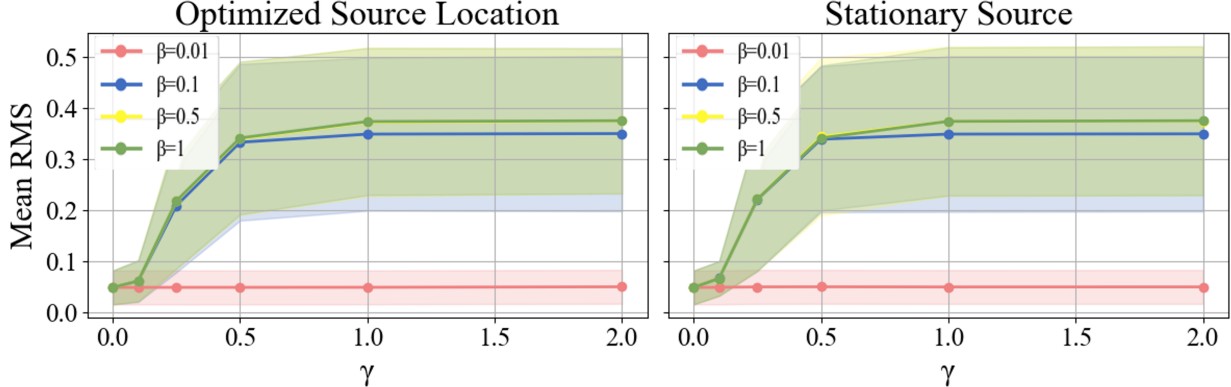

Figure 4: **Mean RMS error with and without source location optimization across varying amplitude ($\beta$) and power ($\gamma$) bounds. Optimizing the perturbation source location yields negligible improvement over a fixed source in the room center.**

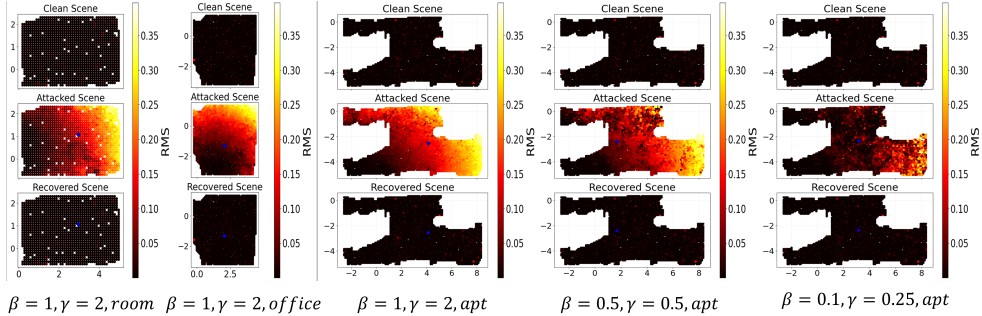

Figure 5: **Spatial error distribution (Mean RMS) for Clean (top), Attacked (middle), and Recovered (bottom) scenarios across three environments (*apt., room, and office*). The "Attacked" row demonstrates that the universal perturbation degrades localization performance uniformly across the entire environment (indicated by high RMS values), rather than exploiting specific positional weaknesses. The "Recovered" row shows that our phase-modulation defense successfully delineates the perturbation, restoring localization accuracy to near-clean levels. Blue markers indicate optimized source locations.**

Figure 5 illustrates spatial RMS error distributions across varying scenes and $\beta, \gamma$ bounds for optimized source location perturbation optimization. We supply similar results for fixed-source location in appendix B. "Clean Scene" denotes non-perturbed localization, "Attacked Scene" depicts localization subjected to the attack from Section 3.3 and "Recovered Scene" shows localization with the perturbation delineation defense methods developed in Section 3.4. Results visually demonstrate the effect of our proposed attack over localization performance, spread across the majority of the environment in all test scenes. The smooth spatial distribution of error across each map implies that our universal adversarial attack method manages to markedly diminish the underlying localization capability throughout the scene. This consistency in error across diverse spatial coordinates implies that the attack successfully generalizes to attack the localization task as a whole, rather than exploiting isolated positional weaknesses. To address any concern of the similarity between fixed and optimized source in Figure 4 stemming from the perturbation source not changing in optimization, we stress that location-optimization experiments do demonstrate some change in source location under optimization. The restricted extent of this change from the center can be explained by the reduced affect of room geometry over perceived perturbation at the sensors as the source is closer to the sensor. This pushes the perturbation

source to remain close to the center during optimization, also insinuating that the conclusion from Figure 4 regarding the effect of source location optimization may change in highly non-convex rooms. We defer exploration of this possibility to future work.

In our setting the exploration of *targeted* adversarial attacks is also of great importance, as such attacks would allow a potential attacker to direct the drone to a specific location within the environment. Our main analysis of the performance of our method in applying targeted attacks is brought in the supplementary (Appendix B.1), however in Figure 6 we present initial results of our developed attack's potential in successful targeted attack. This potential is evident from the presented error heatmap in the attacked scene, depicting RMS error between prediction and adversarial target. For our largest-considered attack bounds, this error is almost zero all along the map. We refer the reader to the supplementary (Appendix B) for further related results, including cases where our targeted attack has proven inefficient.

### 4.3 Defense Results

In this section we analyze the effectiveness of our proposed acoustic perturbation delineation algorithm from Section 3.4. Figure 5 shows qualitative support for our method's capability in drastically reducing adversarial performance decay, producing recovered scenes almost indistinguishable from the corresponding clean scenes - for the most lenient attack constraints, our algorithm reduces localization error from 37% (Figure 4) to below 6%, only marginally higher than the error of 4.87% reported for clean localization. In Appendix E we also report the recovered RMS error statistics, similarly to Figure 4.

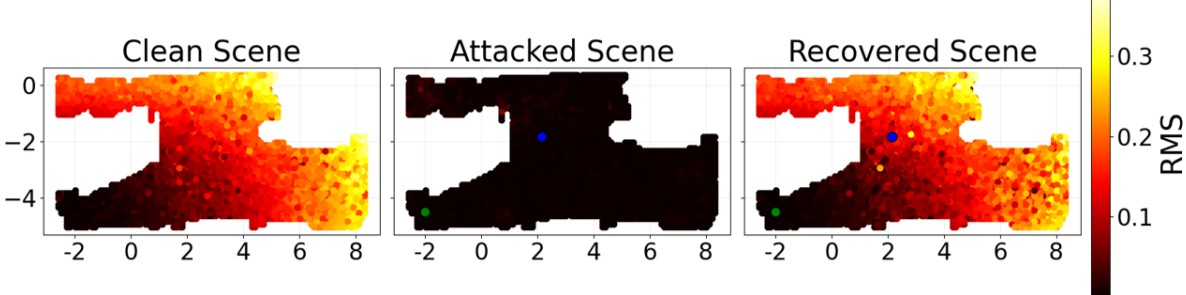

Figure 6: **Targeted attack spatial distribution -** of RMS error relative to the adversarial target (indicated by the blue dot) under a targeted attack with maximal bounds. The uniformly low error (black regions) indicates a highly successful attack, where the drone is consistently deceived into localizing itself at the adversary's chosen coordinates regardless of its actual position in the room.

We attribute the efficacy of our method to the fact that it is capable of retrieving the sensed perturbation with a only single scalar degree-of-freedom, that is the uncertainty in the sampled perturbation value at moment $t = 0$. To further validate our method, in the appendix we provide thorough empirical analysis of this uncertainty, and conclude that this degree of uncertainty has marginal affect over post-reconstruction localization performance.

### 4.4 Noise Modeling

Since we are the first work to address adversarial attacks over acoustic drone localization, in our experiments we wish to assess the impact of acoustic adversarial attacks over drone localization with minimal assumptions over the attacker and the environment. Therefore, in all previous experiments we assume the only sound sources are the drone and the adversarial perturbation. This assumption maximizes the attacker's flexibility and isolates external interference. Nonetheless, in order to provide an initial assessment of our findings in real-world conditions, in this section we test the potential of our developed attacked under the presence of white noise. Table 1 shows final attack accuracy after perturbation optimization. Selected values of $\sigma$ are chosen as std values below of up to 50% of the standard-deviation of the original signal. Notably, the presence of noise does not affect the potential of the attack, in comparison to the clean-setting case.

Table 1: Attack performance under different noise standard-deviation $\sigma$. Rows represent different attack configurations $(\beta, \gamma)$.

| $\beta, \gamma$ / $\sigma$ | 0.01 | 0.025 | 0.05 | 0.075 | 0.1 |
|---|---|---|---|---|---|
| Clean model (under noise) | 0.0492 | 0.062 | 0.081 | 0.116 | 0.147 |
| $\beta = 0.1, \gamma = 0.25$ | 0.22 | 0.226 | 0.23 | 0.26 | 0.261 |
| $\beta = 0.5, \gamma = 0.5$ | 0.345 | 0.339 | 0.327 | 0.346 | 0.342 |
| $\beta = 1, \gamma = 2$ | 0.372 | 0.371 | 0.369 | 0.358 | 0.35 |

## 5  Discussion

In this work, we presented a first study of adversarial attacks on acoustic localization. We formulated a framework for generating universal acoustic perturbations and analyzed their impact on localization accuracy under various conditions. We also introduced a novel algorithm for recovering the adversarial signal. Our attacks increased localization RMS error by up to 37%. Our approach also demonstrated that adversarial optimization of source location has minimal impact on attack effectiveness, enabling efficient attacks with lower computational cost. As a by-product, we trained and evaluated a self-sound-reliant drone localization model on real acoustic data, expanding prior work that focused solely on simulations.

While our work provides a foundational framework for adversarial attacks and defenses in acoustic localization, several limitations remain. First, in this study we focus on 2D acoustic localization. Many real-world localization scenarios incorporate a 6-DoF regression problem, calling for an extension of our contributions to higher-dimension localization. Second, in order to isolate the marginal impact of the adversarial attack, our work assumes a simplistic noise model, that does not grasp the chaotic acoustic nature found in some scenes. Furthermore, while efficient, our perturbation delineation algorithm must be applied for an entire drone cycle for every moment in time $[0, T]$, which requires several seconds to delineate a single perturbative waveform.

Another promising avenue for future research would be the exploration of more-complex adversarial settings, that are not discussed in this work. While our experimental evaluation is focused on a single stationary perturbation source, the proposed defense is theoretically extensible to more-general scenarios. Since acoustic signals in a linear medium superimpose, a multi-source attack effectively manifests as a single complex waveform at the sensor array. Because our defense relies on the independence of external sound sources from the drone's internal phase modulation mechanism, it is potentially capable of extending to any number of stationary, non-adaptive sources without modification. Another viable adversarial setting for future exploration would be active real-time adversarial adaptation done by the attacker. In the case of our proposed defense, it is constrained by causality. The propagation delay of sound and the processing time required to estimate the drone's phase prevent the attacker from reacting instantaneously to the defense mechanism. We hope our findings help pave the way for the development of robust and secure acoustic perception in future autonomous systems.

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

## A  Supplementary Layout

In the following sections we report additional details complementing the main paper, ordered as follows - 1) Section B includes supplementary results for the three main types of experiments conducted in the paper - **Targeted Attacks** - Section B.1 extends figure 6 from the paper, reporting localization performance attained under targeted adversarial attacks, considering a wider range of acoustic scenes and attack bounds, as well as an ablation for adversarial sound source location optimization.
**Source Location Optimization** - Section B.2 provides further comparison between localization capabilities with and without adversarial source location optimization, similar to section 4.2.
$\delta$ **Uncertainty** - Section B.3 elaborates values of location gradients w.r.t sensor-sampled sound (as shown in figure 5) to quantifty the error expected for our perturbation delineation algorithm under the uncertainty in value of the recovered waveform at moment $t = 0$.

we note that our scatter-plot heatmap figures for the *room_2* and *office_4* environments may depict environment shapes slightly different than those known from Chang et al. (2017); Straub et al. (2019) environments. The reason is we do not evaluate localization in locations where the drone cannot be present, due to its simulated physical dimensions from Serussi et al. (2024).

2) Section C provides in-depth implementation, optimization and training-regime details. 3) In Section D we discuss the added computational resources required in adversarial source location optimization, to further stress the importance of our findings from section 4.2, where we claim source-location optimization could prove cost-ineffective in some cases.

4) Section E further provide added details for our proposed perturbation delineation algorithm.

## B  Additional Results

### B.1  Targeted Attacks

A central motivation for the study of adversarial attacks in drone localization is the possibility of a potential attacker directing the drone to a specific location, where it could potentially pose harm to the original user or third-party individuals. We therefore also analyze the effect of targeted adversarial attacks, brought forth in Figure 7. In this figure we show spatial error distributions for targeted attacks in different locations (marked in green on the heatmap, and textually on each sub-figure caption). Note that unlike heatmaps for non-targeted attacks, here the error reported on the heatmap is *RMS from adversarially desired target* (so an all-black heatmap means perfectly successful attack).
We observe that the success of targeted attacks depends on target location. Subfigures *a-c* present very successful adversarial targeted attack, where the adversarially-desired target is predicted almost everywhere in the map. Figures *g-i* present the complementary scenario, where the clean and attacked error maps are nearly identical for all bounds. We speculate this difference stems from the fact that the attack targeted at $[-2, -4.5]$ (namely, the "successful" attack) is focused on a more-likely low-certainty area (given the room

shape in the area of $[-2, 0]$, that is non-representative of the common shape at other scene regions), contrary to the attack at $[8, -4.5]$ (an area where the attack allegedly failed).

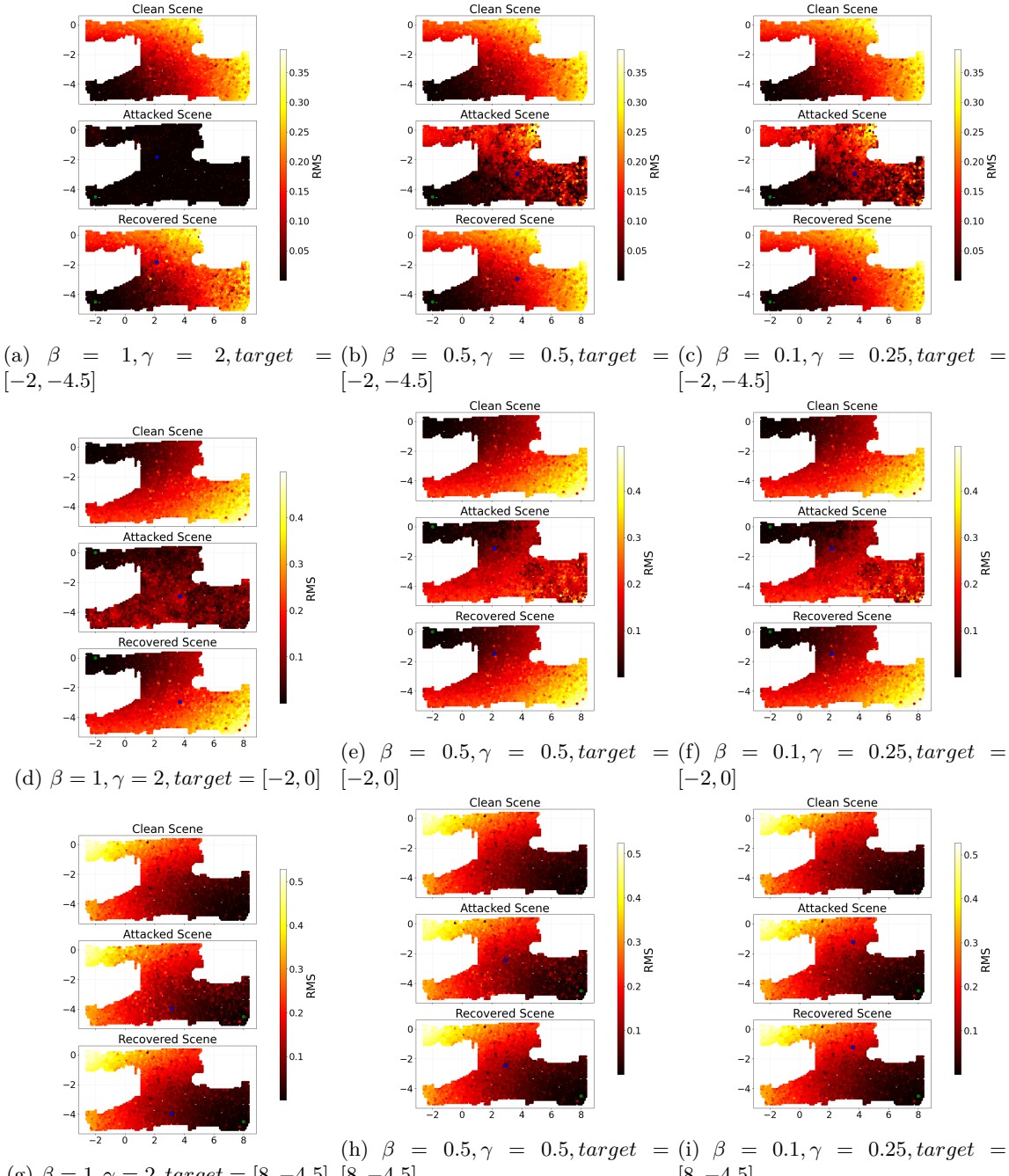

Figure 7: **Clean, perturbed and delineation-recovered spatial error distribution for selected targeted attack bounds** - for selected map targets (in green).

## B.2 Fixed-perturbation source RMS distribution

Figure 8 presents results similar to Figure 5 along with fixed-source location attack optimization. Results support our findings from Section 4.2, stating that location optimization holds limited contribution for attack

success in our tested setting. Similar conclusions can be drawn from results over the office (Figure 9) and room (Figure 10) environments, also complementing the partial results for location optimization in the two latter environments, as shown in Figure 5.

In Figure 11, Figure 12 we present localization RMS mean and standard deviation, similar to the plot from Section 4.2 in the paper. While our conclusion regarding marginal advantage in source location optimization is supported in those plots, we do denote that for these environments, for smaller attack bounds we observe a larger growth in attack efficiency when performing source-location optimization.

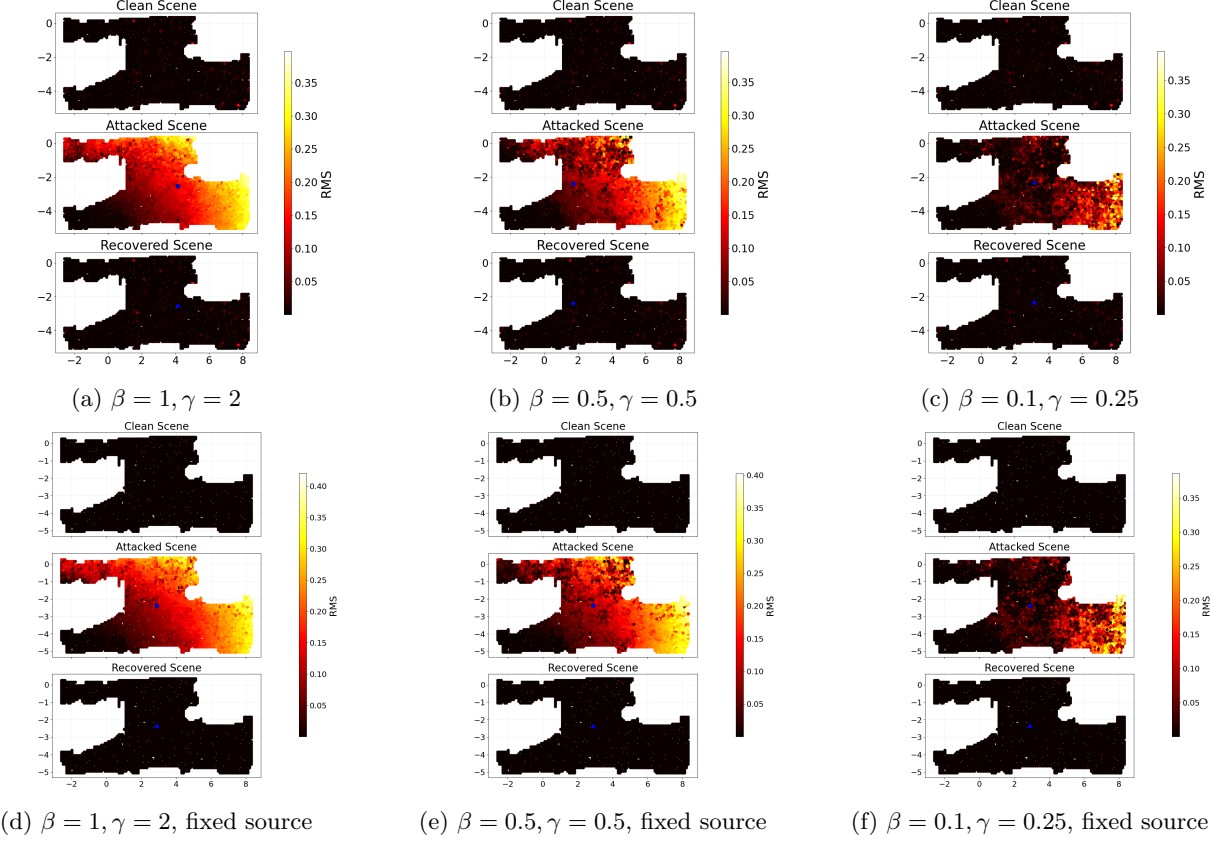

(a) $\beta = 1, \gamma = 2$     (b) $\beta = 0.5, \gamma = 0.5$     (c) $\beta = 0.1, \gamma = 0.25$

(d) $\beta = 1, \gamma = 2$, fixed source     (e) $\beta = 0.5, \gamma = 0.5$, fixed source     (f) $\beta = 0.1, \gamma = 0.25$, fixed source

Figure 8: **Clean, perturbed and delineation-recovered spatial error distribution for selected attack bounds** - for both fixed and optimized source location.

### B.3 Perturbation Delineation Uncertainty

Our perturbation recovery algorithm recovers the exact perturbation waveform up to uncertainty in the sampled perturbation value at moment $t = 0$, in this section we further evaluate our method by assessing the effect of this deviation over model performance. Namely, for a clean input signal $s_{drone}(t)$, our recovered signal in Section 3.4 is $(s_{drone} + \delta)(t)$ for some unknown scalar offset $\delta \in \mathbf{R}$ (that equals the perturbation response at the microphone at moment $t = 0$). Our primary approach for evaluating the repercussion of uncertainty in $\delta$ is by estimation of the spatial distribution of location-wise perturbation values at $t = 0$ (denoted $s'_{\mathrm{p}}(t = 0; \mu)$ and location gradient infinity norms across different sensor locations $\mu$ - $\mathcal{L}_{\infty}(\frac{\partial \mathcal{F}(s_{drone})}{\partial s_{drone}})$.

The product $s'_{\mathrm{p}}(t = 0; \mu) \cdot \mathcal{L}_{\infty}(\frac{\partial \mathcal{F}(s_{drone})}{\partial s_{drone}})$ upper-bounds the location-wise rate of location prediction change as a response of change in $\delta$ exactly equal to our perturbation recovery error $s'_{\mathrm{p}}(t = 0; \mu)$. Results are brought in Figure 13, demonstrating that characteristic values of $s'_{\mathrm{p}}(t = 0; \mu)$ across the map hold negligible response

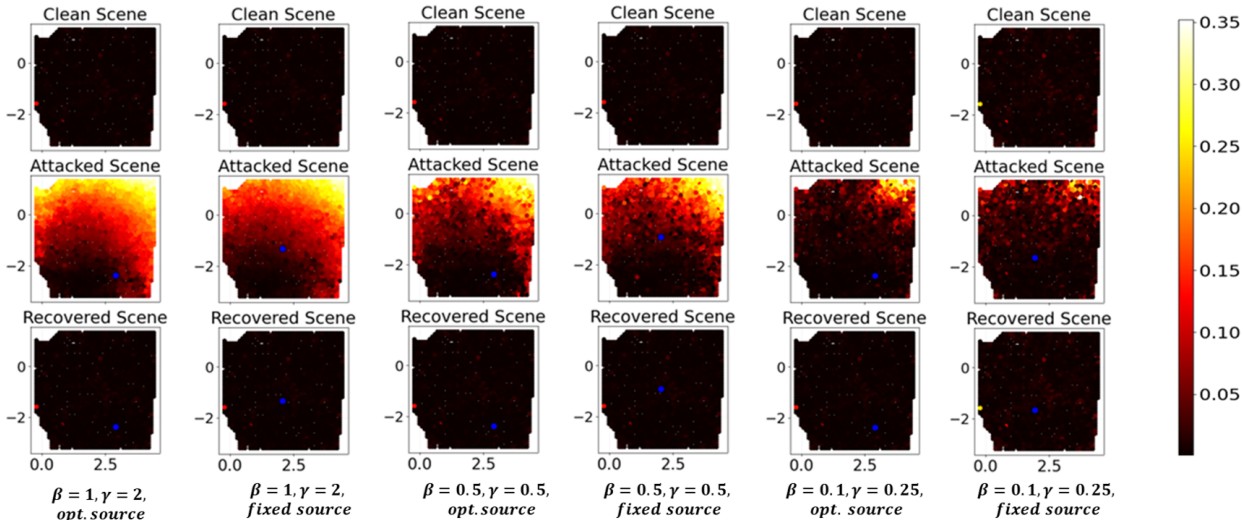

Figure 9: **Office environment spatial error distribution for selected attack bounds** - Office environment.

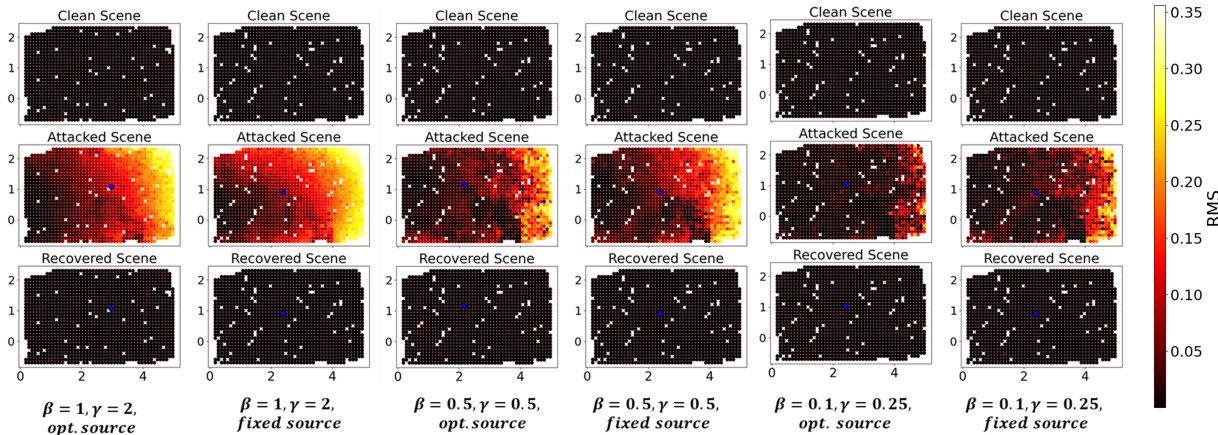

Figure 10: **Office environment spatial error distribution for selected attack bounds** - Room environment.

over location prediction, corroborating the reliability of our delineation method.

To further support our conclusions, we show similar results on the training objective loss gradients, rather than on the location directly. These results are depicted in Figure 14.

# C Training Details

## C.1 Clean Model Training

For training our "clean" localization model (to later be attacked) we train both the forward model and the transformer-encoder inverse model from Serussi et al. (2024)

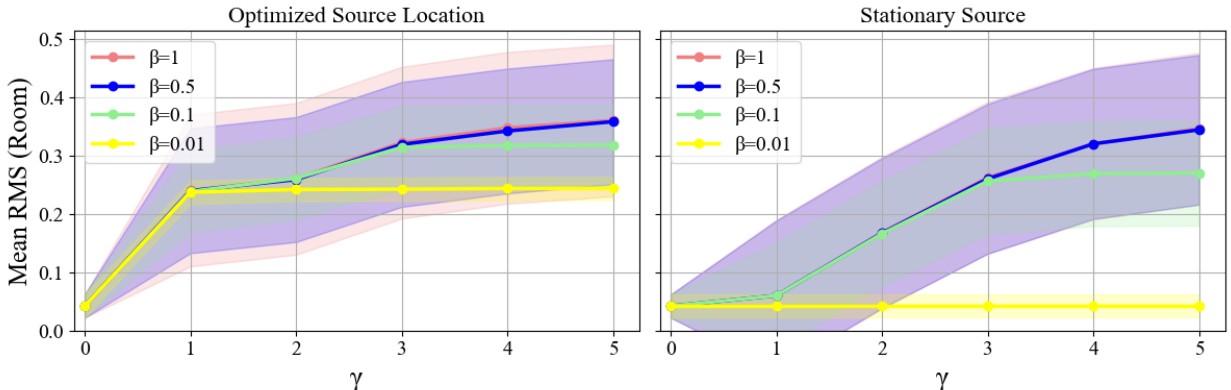

Figure 11: **Mean RMS with and without source location optimization** - Room Environment.

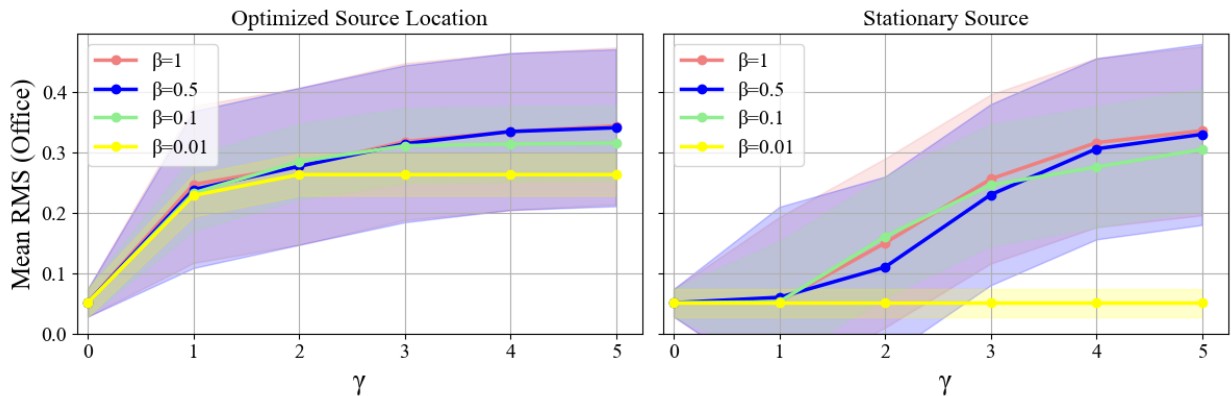

Figure 12: **Mean RMS with and without source location optimization** - Office Environment.

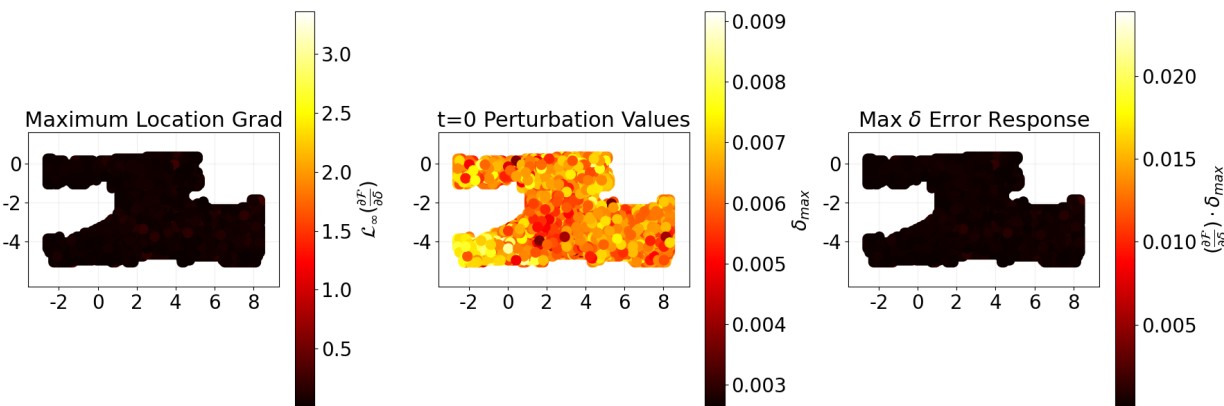

Figure 13: **Spatial distribution of maximum location prediction response to change in $\delta$** - demonstrating responses typically several magnitudes of order lower compared to absolute locations

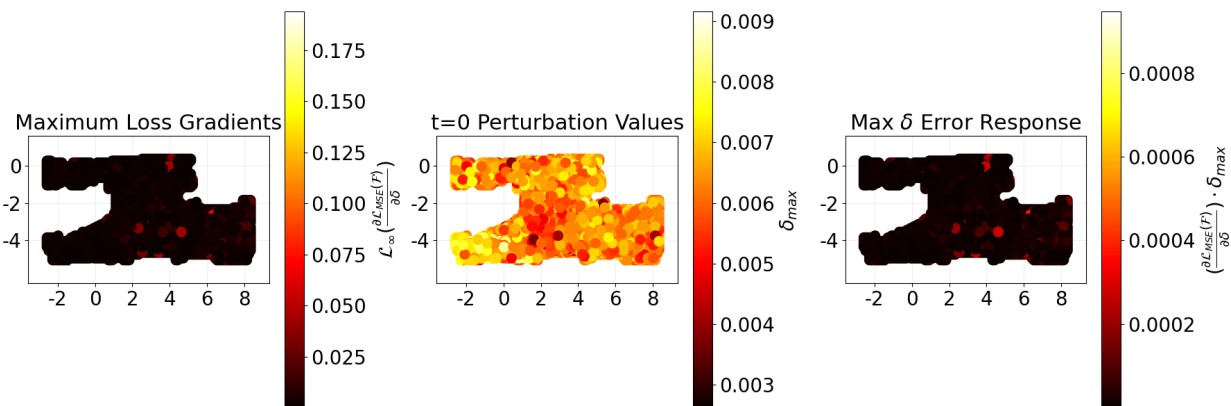

Figure 14: **Spatial distribution of maximum loss response to change in perturbation starting conditions uncertainty** $\delta$.

### C.1.1 Forward Model

We parameterize the self-sound of each rotor with 16 point sound-sources, and optimize using L-BFGS for 50 iterations. The self-sound is then used along with the scene-specific pre-trained RIRs published in Luo et al. (2022) for each test environment.

### C.1.2 Localization Model

We train the localization model with dataset generated by sampling each environment with a $5 \times 5$ cm density. For each such drone location we computed the sound sampled by the microphone array in 32 azimuthal orientations uniformly spaced within the $[0, 2\pi]$ range. With this dataset of sounds sampled by the microphones at different drone locations and orientations ( 125000 samples for the apartment, 65000 for office and 42000 for room) we train the localization model, using the transformer-encoder architecture from Serussi et al. (2024). We subsample each waveform from a temporal dimension of 12000 timesteps to 1200 using a learned linear operator. This subsampled input is then fed into a 3-layered transformer-encoder model with hidden dimension of 1024. This model is trained on batches of 128 for 80 epochs on an NVIDIA RTX-2080 GPU, with phase modulation optimization of 10 sine basis components. Model parameters are optimized with an Adam optimizer, using a learn-rate of 0.00001 for the localization model and a learn rate of 0.05 for the phase modulation parameters.

### C.2 Perturbation Optimization

We train each universal perturbation for 100 PGD iterations, with early-stop of 5 consecutive epochs without loss increase. Each PGD iterations is performed across the entire test set. Minimum frequency is set to $50[Hz]$, maximum to $2000[Hz]$. $\lambda_{amp}, \lambda_{power} and \lambda_{SDF}$ are all set to 1 for all experiments. For location optimization we run all experiments on a NVIDIA A6000 GPU with batch-size of 3. For fixed attack source locations we train on an RTX-2080 with batch-size of 128. Source initial position is set to the center of the environment for all experiments.

## D Resource Analysis

In this section we supply resource consumption details for a single NVIDIA A6000 GPU during a single PGD iteration of perturbation optimization across the entire dataset of 12575 samples, consisting our test set for the *apartment_frl_2* acoustic setting (the largest of the 3 environments we consider). Each table compares iteration time with and without source location optimization. Results show growth in GPU memory consumption by a factor of over 3, and in optimization time growth by a factor of around 2 when applying source location optimization, compared to the fixed-source location counterpart. We further stress

Table 2: RMS Mean $\pm$ Standard Deviation per amplitude ($\beta$) and power ($\gamma$) constraints - after perturbation recovery

| $\beta \setminus \gamma$ | clean | 0.1 | 0.25 | 0.5 | 1 | 2 |
|---|---|---|---|---|---|---|
| **0.01** | $0.0487 \pm 0.0331$ | $0.0568 \pm 0.0331$ | $0.0568 \pm 0.0332$ | $0.0568 \pm 0.0333$ | $0.0568 \pm 0.0332$ | $0.0568 \pm 0.0332$ |
| **0.1** | $0.0487 \pm 0.0331$ | $0.0568 \pm 0.0315$ | $0.0569 \pm 0.0330$ | $0.0570 \pm 0.0334$ | $0.0572 \pm 0.0334$ | $0.0569 \pm 0.0331$ |
| **0.5** | $0.0487 \pm 0.0331$ | $0.0568 \pm 0.0316$ | $0.0569 \pm 0.0330$ | $0.0570 \pm 0.0334$ | $0.0568 \pm 0.0331$ | $0.0570 \pm 0.0331$ |
| **1** | $0.0487 \pm 0.0331$ | $0.0568 \pm 0.0316$ | $0.0570 \pm 0.0331$ | $0.0571 \pm 0.0333$ | $0.0572 \pm 0.0332$ | $0.0573 \pm 0.0333$ |

that while, for the sake of comparability, we only analyze for low batch-sizes of up to 3 (for which we are able to fit optimization in memory using a single GPU in both cases), fixed-location optimization can be expedited by increasing the batch-size.

Table 3: Single PGD iteration memory consumption across batch sizes

| Batch Size | Optimized Location (GB) | Fixed Location (GB) |
|---|---|---|
| 1 | 15.850 | 5.562 |
| 2 | 29.746 | 9.184 |
| 3 | 43.542 | 12.078 |

Table 4: Single PGD iteration runtime across batch sizes

| Batch Size | Optimized Location (seconds) | Fixed Location (seconds) |
|---|---|---|
| 1 | 7592 | 3979 |
| 2 | 6615 | 3491 |
| 3 | 6322 | 3201 |

we note that the batch-size of 128 mentioned in Section C is used by internally batching RIR computations to batches of 2, and only inserting the batch-size of 128 onto the localization model.

# E    Phase Modulation Perturbation Delineation

In this section we provide further evidence for the prowess of our adversarial perturbation recovery algorithm from Section 3.4, added to figures 13,5.

In table 2 we report RMS error mean and standard-deviation for all amplitude and power bounds evaluated in Section 4.3, similarly to results shown in Section 4.2. We observe marginal performance differences between clean and recovered perturbation performance.

