# OpenReview forum: "On Adversarial Attacks In Acoustic Localization"
_TMLR — Accepted by TMLR_

### Review · Reviewer_Nwh5 · 2025-12-18

**Summary Of Contributions:**

Contributions:

1. Problem Formulation: Provide the first systematic study of adversarial attacks on acoustic-based drone localization relying on self-generated propulsion noise.

2. Attack Framework: Formulate a fully differentiable, white-box universal acoustic attack from an external stationary sound source under realistic frequency and amplitude constraints.

3. Defense Method: Propose a novel test-time defense based on active rotor phase modulation that enables separation and removal of adversarial perturbations without explicit attack detection.

Key Strengths:

1. The paper addresses a novel and previously unexplored security vulnerability in acoustic localization.

2. The proposed defense is physically grounded, elegant, and does not rely on heuristic detection thresholds.

Weaknesses:

1. The attacker model (in Eq. (4) ) focuses on a single stationary, non-adaptive sound source; further discussion on how the proposed defense may extend to more adaptive or multi-source adversaries would strengthen the scope of the work.

2. While the evaluation uses realistic acoustic modeling, all experiments are conducted in simulation; additional discussion of the practical feasibility of rotor phase modulation on real drone platforms would improve clarity regarding real-world deployment.

**Audience:**

Yes

**Audience Explanation:**

Acoustic sensing and robust perception are relevant to many robotics and autonomous systems, particularly in GPS-denied or visually degraded environments. The paper introduces an adversarial threat model and a principled defense mechanism that are likely to be of interest to researchers working on robustness, sensor security, and embodied AI within the TMLR community.

**Broader Impact Concerns:**

None.

**Claims And Evidence:**

Yes

**Claims Explanation:**

The paper provides clear empirical evidence that acoustic localization models are vulnerable to adversarial sound perturbations. Consistent degradation is observed across multiple environments and attack settings. The proposed defense is evaluated under the same conditions and recovers localization performance close to clean baselines, supporting the main claims of the paper.

**Requested Changes:**

1. Clarify the scope of the attacker model and discuss how the proposed defense may extend to more adaptive or multi-source adversaries. More specifically, the attacker model considered in Eq. (4) focuses on a single stationary, non-adaptive perturbation source. Additional discussion clarifying this scope, as well as potential extensions of the proposed defense to more adaptive or multi-source adversarial settings, would improve the clarity and generality of the contribution.

2. Further discuss the practical feasibility of phase modulation in real-world drone platforms.

---

### Review · Reviewer_TQsR · 2025-12-18

**Summary Of Contributions:**

This work aims to analyze possible adversarial perturbations that can interfere with acoustic localization in drones. It also proposes phase modulation as an effective defence against such adversarial attacks.

**Strengths**:

- Using NAFs instead of compute-heavy ISMs for RIR computation would be a strong contribution to the literature on acoustic localization since it brings down the cost of training the localization model.

- The attack is universal and does not rely on exploiting positional weaknesses of certain locations. It affects localization uniformly across the space.

- The attack is affected minimally with source location optimisation demonstrating robustness in attack efficacy with a fixed perturbation source.

**Weaknesses**:

1. Author(s) modify Serussi et al.(2025)'s approach to acoustic localization replacing ISMs with NAFs, but do not provide any details about how the new formulation varies in efficiency compared to the baseline (Serussi et al. (2025)). While they do say these modifications fall outside the main contributions of the paper, a comparison of the 2 approaches would ensure that this new formulation does not necessarily cause performance degradation.

2. In Section 4.1, the author(s) mention using pretrained NAFs for their experiments. Following that, it says that the environments attacked in the paper are a subset of the NAF's pertaining data. Can the author(s) comment on whether the results will be affected if the pretraining and attack environments differ in distribution?

3. In Section 4.1, the author(s) state, "Nonetheless, we emphasise that our developed attack and defence methods apply to any combination of drone configuration and acoustic localization algorithm." This claim is not accompanied by any supportive evidence.

4. (Minor) The attack is white-box, and as such, it relies on certain assumptions about the adversary's access to the localization model. The author(s) will benefit by detailing these assumptions in the paper.

5. (Minor) In Section 3.3.4, it says, "perturbation source location $x_{\mu}$". Should it not be $x_{p}$, the perturbation source location?

**Audience:**

Yes

**Audience Explanation:**

Acoustic localization is proposed as a computationally efficient alternative to vision-based localization. Studying possible interference in localisation by adversarial perturbations is relevant to the research towards robust acoustic localisation algorithms.

**Claims And Evidence:**

No

**Claims Explanation:**

Weaknesses #1, 2 and 3 undermine the paper's acceptance. If the author(s) could address them, I am amenable to revising my stance.

**Requested Changes:**

It would greatly strengthen the paper if the author(s) could address the list of weaknesses/ modify the submission accordingly.

Additionally, the presentation of the paper would benefit from,
- Using capitalised references to figures and sections, i.e., using "Figure 6" instead of "figure 6" to refer to them in the text. This can be done by using "\Cref" in place of "\cref" from the cleveref package.
- In places where the results are present in the supplementary materials, specify references to the sections in the Appendix about them instead of directing readers to go on a search in the supplementary section.
- Captions in figures are supposed to be a standalone description of the message authors are trying to convey. The bare minimum description expects the reader(s) to jump back and forth between the text and the figure to get an idea of what the intended message is. Complement the text by captions detailing the experimental setting + primary observation from the figure.

---

### Review · Reviewer_m4Tc · 2025-12-27

**Summary Of Contributions:**

This paper presents the first systematic study of adversarial attacks and defenses for acoustic-based drone localization relying on self-generated propulsion noise. The authors formulate a physically realistic threat model where external sound sources disrupt localization, utilizing Neural Acoustic Fields for gradient-based attack optimization. Experiments demonstrate that universal acoustic perturbations significantly degrade accuracy, increasing RMS error from under 5% to over 30%.

**Audience:**

Yes

**Audience Explanation:**

Studying adversarial attack in acoustic localization is an important topic for machine learning application.

**Broader Impact Concerns:**

N.A.

**Claims And Evidence:**

Yes

**Claims Explanation:**

1. This paper addresses the adversarial robustness of acoustic-based drone localization, a topic that has received little attention compared to vision- or LiDAR-based perception systems.
2. The integration of Neural Acoustic Fields enables a fully differentiable forward model, allowing gradient-based optimization of adversarial waveforms and source locations.
3. The proposed defense based on rotor phase modulation is conceptually elegant.

**Requested Changes:**

1. In most experiments, the setup assumes zero interference from other acoustic signals aside from the drone and the attacker. While a noise analysis is provided in Section 4.4, the primary evaluation environment is somewhat idealized compared to chaotic real-world acoustic environments with multiple unrelated noise sources.
2. The threat model focuses on a stationary, omni-directional sound source. While this is a necessary starting point, it does not account for mobile attackers or directional interference, which might present different challenges for the proposed phase-modulation defense。
3. Please replace with high-resolution figures.

---

> ### Author Response · Authors · 2025-12-27
>
> We thank reviewer m4Tc for reviewing our work. We wish to address the weaknesses raised:
> 1. Idealized Evaluation - We agree that our set of experiments does not cover the common realistic setting, and that this is an important future direction for exploration. Our motivation for this is that we wish to isolate the variables to measure the marginal impact of the adversarial attack. If the environment is too noisy, it masks the specific effect of the gradient-based perturbation. Section 4.4 serves as the bridge to reality, however creating an actually representative and realistic noise model would be challenging due to the chaotic and high-variance noise found in actual scenes. We therefore limit our work focus on an initial theoretical framework. We've uploaded a revised version addressing this limitation in section 5.
>
> 2. Limited variety of adversarial source model - we agree with this point as well. We believe directed sources would be an interesting experiment, however we also emphasize that theoretically we anticipate similar behavior (error distributions after attack and defense) in the directed case - as long as there is some scatter of sound waves decaying through time, our defense should be able to filter the attack. For a mobile attacker, we agree our current formulation does not fully capture this scenario. Exploration of such sources would require the consideration of additional physical attributes of sound propagation (e.g. Doppler) and in diverted to future work.
>
> 3. Regarding figure resolution - we kindly request the reviewer to point us towards which figures seem problematic in resolution, and we will willingly correct any issues with presentation. If the reviewer's intention is to the several scatterplots included in the paper - we intentionally use large scatterpoints so the spatial distribution of error can be seen clearly.

---

### Decision · Action_Editor_imSX · 2026-02-14

**Recommendation:** Accept as is

**Additional Comments:**

This submission makes a well-executed contribution by initiating the study of adversarial robustness in acoustic drone localization. The attack formulation is principled and leverages differentiable acoustic modeling. The defense is physically grounded and deployable in principle on standard multi-rotor platforms. The evaluation is comprehensive within the simulated setting and clearly supports the claims.

The paper is appropriately scoped, transparent about its limitations and outlines concrete directions for future work. Given the reviewers’ consensus and the satisfactory revision, the paper meets TMLR’s criteria for soundness, clarity, and relevance.

**Audience:**

Yes

**Audience Explanation:**

The paper targets robustness and security of learned perception systems, specifically in acoustic sensing for drone localization. Researchers working on adversarial robustness, embodied AI, robotics, acoustic modeling, and sensor security would find the findings relevant. While the application domain is specialized, the threat modeling and defense methodology are of broader interest to the TMLR community.

**Claims And Evidence:**

Yes

**Claims Explanation:**

This paper provides the first systematic study of adversarial attacks on acoustic-based drone localization.

All three reviewers agree that the main claims are supported by clear empirical evidence. The paper formulates a well-defined attacker model and demonstrates consistent performance degradation (e.g., large increases in RMS localization error) across multiple environments.

Concerns raised during review were addressed in the revision including clarifications of assumptions, corrected notation, improved figure references/captions, and an expanded section on limitations and future work. The reviewers confirmed that these revisions satisfactorily addressed their major concerns.